# Automatic Illumination Control Method for Indoor Luminaires Based on Multichromatic Quantum Dot Light-Emitting Diodes

**DOI:** 10.3390/mi13101767

**Published:** 2022-10-18

**Authors:** Hua Xiao, Guancheng Wang, Wenda Zhang, Sirong Lu, Bingxin Zhao, Zhanlang Wang, Yanglie Li, Jiada Liu

**Affiliations:** 1School of Electronic and Information Engineering, Guangdong Ocean University, Zhanjiang 524088, China; 2Technology Development Centre, Shenzhen Institute of Guangdong Ocean University, Shenzhen 518120, China; 3Department of Electrical and Electronic Engineering, Southern University of Science and Technology, Shenzhen 518055, China; 4Shenzhen Institute for Quantum Science and Engineering, Southern University of Science and Technology, Shenzhen 518055, China; 5The Theory Technology Co., Ltd., Shenzhen 518126, China

**Keywords:** white light-emitting diode, energy saving, illuminance control, spectral optimization

## Abstract

Energy saving and visual comfort are two main considerations in designing of automatic illumination control systems. However, energy-saving-oriented illumination control always causes optical spectra drifting in light-conversion-material-based white light-emitting diodes (WLEDs), which are conventionally used as artificial luminaires in indoor areas. In this study, we propose a method for InP quantum dot (QD)-based WLEDs to minimize optical energy consumption by considering the influence caused by the outdoor environment and neighboring WLED units. Factors of (a) dimensions of room window and WLED matrix, (b) distance between WLED units, lighting height, species of InP QDs, and (c) user distribution are taken into consideration in calculation. Parameters of correlated color temperature (CCT) and color rendering index (Ra) of the WLED matrix are optimized according to the lighting environment to improve user visual comfort level. By dynamically controlling the light ingredients and optical power of WLEDs, we optimize the received illuminance distribution of table tops, improve the lighting homogeneity of all users, and guarantee the lowest energy consumption of the WLED matrix. The proposed approach can be flexibly applied in large-scale WLED intelligent controlling systems for industrial workshops and office buildings.

## 1. Introduction

Light-emitting diodes (LEDs) have emerged as the fourth-generation solid-state lighting devices over the last few decades due to their advantages of long lifespan, high efficiency, reliability, and eco-friendliness [1,2,3,4]. They have been widely applied in numerous scenarios, such as residential and outdoor illumination [5], rehabilitation therapy [6], visible light communication [7], agricultural planting [8], and indoor localization [9]. Among these applications, residential illumination shows high energy consumption, which reaches approximately 20–40% of the total annual energy consumption of buildings [10]. Two main approaches have been proposed to decrease this spending; the first is to reduce light utilization by lowering the dimming level, the other is to improve the light conversion efficiency of LEDs. Additional to LED arrangement optimization [11], daylight energy has been used in hybrid illumination systems to reduce optical power of artificial lamps, which helps to reduce the energy consumption by at least 10% [12]. With these approaches, LED-equipped greenhouses can effectively achieve the optimal lighting status, and can save electricity costs of up to 45% per year [13].

In the energy-saving-oriented illumination control process, light quality and visual comfort are the two main considerations [14]. Light quality is critical for user visual comfort, sleep quality, and working efficiency. Visual comfort represents the subjective reaction of users to the quantity and the quality of light, and can be determined by the trade-off effects between optical spectra, luminance uniformity, color rendering, and light intensity of white LEDs (WLEDs). To guarantee user visual comfort, the Illuminating Engineering Society (IES) has developed light level recommendations for different scenarios. In a working environment, a minimum light level of 300 lux and a maximum light level of 500 lux are recommended to guarantee user visual comfort [15].

To maximize energy saving efficiency and user visual comfort, researchers have conducted numerous studies. K. R. Wagiman et al. [16] reported that visual comfort is influenced by the dimming level of LEDs, and proposed the concept of the illuminance uniformity deviation index (IUDI), which simultaneously considers the luminance uniformity, visual comfort, and energy produced by daylight and artificial LED lamps. A. Ikuzwe et al. [15] formulated a) a luminous flux degradation model that considers the users’ lighting level requirements and b) an energy maintenance optimization model that considers the luminous flux degradation. Factors such as the optimal number of lamps to be replaced, maintenance schedules, and brightness dimming level were all taken into consideration. To fulfill the photochemical requirements of the photocatalytic reactor, T. Tapia-Tlatelpa et al. [17] considered the LED species and the amount of light, the optimum distance between LEDs, as well as LED arrays with different topological structures, and proposed a methodology to optimize the distribution of LED arrays and their energy consumption. E. N. D. Madias et al. [18] proposed a multiobjective optimization model for artificial luminaires to minimize energy consumption, maximize lighting uniformity, and maintain the illuminance at an appropriate level. In their case study, 15 LED luminaires were used to realize a significant energy saving percentage of approximately 20%. Among these studies, methods such as the multiobjective particle swarm optimization algorithm [16], nondominated sorting genetic algorithm [14], generalized extremal optimization algorithm [19], neural network algorithm [20], and multiobjective evolutionary algorithm [17] have been proposed to optimize parameters such as dimming level, illuminance, and light spatial uniformity.

Three unsolved problems exist among these illumination control studies. Firstly, these studies seldom simultaneously consider time-varying natural sunlight penetrating through the window and optical power originating from neighboring WLED units, such as the sunlight-related studies of [12,13]. Most of them only consider using natural sunlight to reduce illumination electricity; however, the amount of optical energy contributed by neighboring WLEDs shows a strong influence on the illumination intensity and uniformity of indoor areas. Secondly, for light-conversion-material-based WLEDs, optical spectra of WLEDs are controlled by injection currents of LED chips or by PWM modulation. When the injection current or the duty cycle of the current pulse of the blue LED chip increases, the amount of blue light increases, which will directly influence the proportion of light in long wavebands. Both methods can induce the floating of spectra parameters, such as correlated color temperature (CCT) and color rendering index (Ra), resulting in the mismatch between real illumination parameters and target illumination parameters. Thirdly, most studies only optimize the optical power emitted by WLEDs; few of them consider the optical power received by table tops under the WLED matrix. Therefore, it is challenging to simultaneously guarantee the table top brightness and illuminance uniformity, optimize optical spectra of WLEDs, minimize the energy expenditure, and satisfy the illumination standard.

In this study, we aim to solve these problems with an intelligent controlling algorithm. Natural light penetrating through the window, light emitted by neighboring WLEDs, and user distribution of the room are considered in our study to find the optimal illuminating scheme for the WLED system under different circumstances. Additionally, the proposed method also provides a feasible approach to modulate light-conversion-material-based WLEDs, thus optimizing their optical spectra, CCT, and Ra values.

## 2. Monochromatic Spectra Preparation

We select InP quantum dots (QDs) with different peak wavelengths as light conversion materials to produce white light, since InP QDs provide high stability, wide optical spectra for illumination, high wavelength tunability from blue color to near-infrared, and eco-friendliness in comparison with conventional cadmium-based QDs. The improvement of luminance, lifetime, quantum efficacy, and synthesizing methodology of InP QDs in recent studies point out the huge potential of InP QDs in illumination and displaying [21,22]. The synthesizing steps of InP QDs used in this study are given in [23]. As shown in Figure 1, we select the spectra of InP QDs with peak wavelengths of 513 nm, 527 nm, 556 nm, and 600 nm, respectively (measured by Hamamatsu Quantaurus-QY, model no. C11347). Blue LED chips (302 × 198 µm^2^, 460 nm, Hualian Co., Ltd., Xiamen, China) are selected as excitation sources for InP QDs. Blue LEDs provide a narrow full width at half maximum (FWHM) of 33 nm, while InP QDs show wider FWHM of approximately 55–75 nm. Here, we assume that peak wavelengths and FWHM of monochromatic spectra are stable under different excitation light intensities.

Since it is unavoidable to cause shifts in CCT and Ra values when we adjust the light intensity of commercial WLEDs, we strategically adopt a WLED structure that consists of three blue LED chips and two species of InP QD materials. As shown in Figure 2, two blue LED chips are respectively packaged with InP QD films with different peak wavelengths. The other naked blue LED chip is placed to provide supplementary blue light for the trichromatic WLED. Thus, the optical spectra of this WLED system possesses three humps, and can be dynamically adjusted by controlling the driving currents of three independent blue LED chips (*I_i_*, *I_j_*, and *I_k_*) to obtain different CCT values. Here, we give the calculation method of *I_i_*, *I_j_*, and *I_k_* as follows.

To facilitate the study, normalized monochromatic spectra of blue LED chips and two species of InP QDs can be referred to as *S_B_*(*λ*), *S_Q_*_1_(*λ*), and *S_Q_*_2_(*λ*), where *λ* is the light wavelength, *S_B_*(*λ*) is the spectrum of three blue LED chips, *S_Q_*_1_(*λ*) and *S_Q_*_2_(*λ*) are the spectra of two species of InP QD materials, respectively; *S_B_*(*λ*) and *S_Q_*_1_(*λ*) are combined as a two-hump WLED system, as well as *S_B_*(*λ*) and *S_Q_*_2_(*λ*). With this method, we are capable of assembling a three-hump WLED system, which provides wide spectra and high color rendering capability. The normalized spectra of mixed white light *S_W_*(*λ*) can be described as a nonlinear combination of *S_B_*(*λ*), *S_Q_*_1_(*λ*), and *S_Q_*_2_(*λ*), respectively, as follows.
(1)$$S_{W}\left(\lambda \right)=A_{i}\left(I_{i}\right)S_{B}\left(\lambda \right)+A_{j}\left(I_{j}\right)\left(S_{B}\left(\lambda \right)+\alpha \left(I_{j}\right)S_{Q1}\left(\lambda \right)\right)+A_{k}\left(I_{k}\right)\left(S_{B}\left(\lambda \right)+\beta \left(I_{k}\right)S_{Q2}\left(\lambda \right)\right),$$
where *A_i_*(*I_i_*), *A_j_*(*I_j_*), and *A_k_*(*I_k_*) are proportionality coefficients of the blue LED and two species of InP QDs, and they are controlled by independent driving currents (*I_i_*, *I_j_*, and *I_k_*) of three blue LED chips; *α*(*I_j_*) and *β*(*I_k_*) are proportionality coefficients of light produced by the blue LED chip and the corresponding InP QD film, and are determined by the design of the whole WLED system. The nonlinear relationship between *α*(*I_j_*), *β*(*I_k_*), and the driving currents of the blue LED chips, can be measured while the packaging structure of the InP-QD-based WLED is confirmed. Linear interpolation method can be applied to obtain values of *α*(*I_j_*) and *β*(*I_k_*) under driving currents of *I_j_* and *I_k_*. Thus, we only need to independently control *I_i_*, *I_j_*, and *I_k_* in order to optimize the multichromatic optical spectrum, CCT, and Ra values, as well as minimize the energy consumption of the WLED system.

To guarantee the successful conducting of the above method, InP QD films should be thick enough to guarantee the adequate amount of light in long waveband. Additionally, the error, caused by WLED degradation, in the calculation results of *I_i_*, *I_j_*, and *I_k_* can be solved by adding calibration parameters before *I_i_*, *I_j_*, and *I_k_* according to the degradation standard of the WLED. In comparison with (1) the single LED chip and multi-QD film-based WLEDs and (2) RGB WLEDs, merits of our method include: (1) similarity and high controllability of the driving systems of three blue LED chips; (2) high flexibility in multichromatic spectra regulation; (3) wide spectra over visible light area.

## 3. Calculation of Angular Illuminance Distribution of WLEDs

Before calculation, we set the plane of table tops as the receiving plane of our model, assuming (a) the positions of WLEDs are symmetrically distributed with room size, (b) the dimension of a WLED is much smaller than the illumination distance of the room, and the WLED can be regarded as a point light source. Two key problems should be solved to realize automatic illumination control. The first is to calculate the illuminance received per unit area of table tops under different WLEDs, the other is to modulate the light emission of WLEDs to guarantee that all illuminated table tops have uniform optical power distribution and provide similar chromatic performance. To solve these problems, we calculate the illuminance of each table top superimposed by the outdoor natural light and the light from neighboring WLEDs. Regarding the illuminance received by the unit area of the table top as the illuminance of the table top, the received illuminance produced by the corresponding WLED in the top position and by neighboring WLEDs can be defined as *E_θ_*_=0_ and *E_θ_*_≠0_, respectively, as shown in Figure 3a. The calculation steps of *E_θ_*_=0_ and *E_θ_*_≠0_ are presented as follows.

The optical power of the WLED is distributed into the interspace within the spatial emitting angle, which is determined by the optical design of the lampshade. As shown in Figure 3b, for an illuminated arc surface that has a distance *R* from the WLED, we can mathematically divide this surface into numerous small circles. For each circle, the received optical power distribution from the WLED is uniform in all directions. For the circle with a radius of r, the width of the circle equals *dr*, where *dr* = *Rdθ*, Thus, the area of the small circle (*dS_circle_*) can be described as
(2)$$dS_{circle}=2\pi rdr=2\pi R^{2}sin\theta d\theta .$$

We assume that the light distribution of WLEDs is a standard normal distribution. Therefore, the luminance flux per unit area of the arc surface can be expressed as $\tau e^{-r^{2}}$, where τ is the constant to be calculated. Supposing the total emission luminance flux of the WLED equals *Φ*_1_, and the distribution angle of WLED is *2θ*′, we can describe the integral of total optical power received by the arc surface area as
(3)$$\Phi_{1}=\int dS_{circle}\cdot \tau e^{-r^{2}}=\int_{0}^{\theta^{\prime }}2\pi R^{2}sin\theta d\theta \cdot \tau e^{-(R^{2}sin^{2}\theta )}.$$

The above function can be calculated as
(4)$$\Phi_{1}=-2\pi R\tau e^{-R^{2}}\int_{0}^{\theta^{\prime }}e^{R^{2}cos^{2}\theta }dRcos\theta ,\;\;$$
where *Rcosθ* can be defined as *u*. Therefore, Equation (4) can be described as
(5)$$\Phi_{1}=-2\pi R\tau e^{-R^{2}}\int_{R}^{Rcos\theta^{\prime }}e^{u^{2}}du.$$

Thereby,
(6)$$\tau =\frac{\Phi_{1}}{-2\pi R\tau e^{-R^{2}}\int_{R}^{Rcos\theta^{\prime }}e^{u^{2}}du}.$$

Illuminance (*E*) is defined as the luminance flux per unit area, and can be described as
(7)$$E=\tau e^{-R^{2}sin^{2}\theta }.$$

Equation (7) demonstrates that *E* is determined by *R* and *θ*. As shown in Figure 3a, for a unit area that presents an included angle with the normal line of the WLED emission surface, the distance between the corresponding arc surface and the WLED can be described as *R*′, where *R*′ = *R*/*cosθ*. Therefore, $E_{\theta =0}=\tau$ and $E_{\theta \neq 0}=\tau e^{-R^{2}\theta^{2}}$.

## 4. Algorithm Design

Figure 4 illustrates the implementation steps of the proposed automatic illumination control program. First, we initialize the procedure and load the original data, including the number of WLED units, the window dimension and position, the dimension of the WLED matrix, the geometrical structure of the WLED matrix, the local time, monochromatic spectra of the LED chip, and light conversion materials. Afterwards, we select the target CCT value according to the local time and outdoor illumination environment to optimize the optical spectrum of WLEDs and obtain the optimal Ra value. For example, if the sunlight is strong, we can set the target CCT to around 6000 K, which is a medium CCT value to improve user working efficiency in the daytime. However, if the sunlight is weak or non-existent, we can set the target CCT around 4000 K, which can provide a warm and comfortable feeling for users. All these parameters are adjustable according to the users’ lighting preferences. After we automatically obtain the target CCT value, spectral optimization can be conducted according to the method described in [24]. The optimization aim is to obtain an optimized spectral power distribution (*Φ*_0_(*λ*)) that provides the optimal Ra under the target CCT.

Since the numerical value of *Φ*_0_ reflects the radiant power of the WLED, the relationship between luminance flux (*Φ*_1_) and *Φ*_0_ can be calculated as [25]
(8)$$\Phi_{1}=683\int_{380\;}^{780\;}\Phi_{0}\cdot V\left(\lambda \right)d\lambda$$
where *V*(*λ*) is the human eye sensitivity function.

During the calculation process, we mainly focus on three parameters expressed in matrix form. The first is the target illuminance distribution of table tops (*E_t_*), which is influenced by the lighting WLED and its neighboring WLEDs. The second is the optimized illuminance distribution of table tops (*E_o_*), which is mainly influenced by the WLED, its neighboring WLEDs, and natural light from the window. The third is the spatial illuminance distribution of the WLED matrix (*E_r_*), which is the real illuminance produced by WLEDs without any influencing factors. For a specific illuminated table top under the WLED, the total amount of illuminance received from neighboring WLED units is defined as $e_{nei}=\sum_{k=1}^{k=\varepsilon }e_{k}$, where ɛ is the number of neighboring WLEDs, $e_{k}$ is the illuminance produced by the neighboring WLED. *E_nei_* is defined as the matrix of $e_{nei}$ for the WLED matrix. For all illuminated table tops, the illuminance received from the window is defined as $E_{win}\left(\theta^{\prime \prime }\right)$, where $\theta^{\prime \prime }$ represents the included angle between a) the connection line from center point of the window to the illuminated table top and b) the normal direction of the window. $E_{win}\left(\theta^{\prime \prime }\right)$∈*R^m^*^×*n*^, where m and n are the row and column numbers of the WLED matrix, similar to *E_t_*, *E_o_*, *E_r_*, and *E_nei_*. Here, we define the elements in *E_t_*, *E_o_*, *E_r_*, and $E_{win}\left(\theta^{\prime \prime }\right)$ as $e_{t}$, $e_{o}$, $e_{r}$, and $e_{win}\left(\theta^{\prime \prime }\right)$, respectively. The number of elements corresponds to the number of WLEDs.

For each illuminated table top, we regard the initial illuminance value of $e_{t}$ = $e_{r}$+$e_{nei}$ as a fixed illuminance value for all illuminated table tops to achieve. The optimized illuminance distribution of all table tops can be described as $e_{o}$ = $e_{r}$*+*$e_{win}\left(\theta^{\prime \prime }\right)$+$e_{nei}$. The key to eliminating the numerical difference between $e_{o}$ and $e_{t}$ is to dynamically adjust *Φ*_0_ of all WLEDs. Here, we set the varying step of *Φ*_0_ as 0.1 W, while we adjust *Φ*_0_, and observe the variation in *E_o_*. If the difference between $e_{o}$ and $e_{t}$ is decreasing and within the error range (*δ*), the iteration stops for this specific WLED. Afterwards, iteration steps are conducted for all WLED units of the matrix. We assume that all users are using WLEDs at the beginning, and some WLED units are turned off according to user demand, and we can reconduct the above steps to obtain a new balance between all lighting WLED units.

With our method, the optimized *E_t_*, *E_o_*, and *E_r_* for the WLED units that meet the requirements are finally obtained for comparison. Other parameters, such as *I_i_*, *I_j_*, and *I_k_*, can be calculated according the relationship between the optical distribution of different WLED structures, the numerical value of *Φ*_0_, and parameters of *A_i_*(*I_i_*), *A_j_*(*I_j_*), and *A_k_*(*I_k_*) if necessary.

## 5. Results and Discussion

Figure 5 illustrates three different scenarios of the indoor area with target CCT value of 3000 K, 4000 K, and 5000 K, respectively. To achieve target CCTs, we use the combination of humps 1, 3, and 4 (Figure 1) to shape spectra with wide and uniform distribution. Thus, we only adjust the emission proportion of different humps so we can control the shape of the optical spectrum. Optimized optical spectra of the WLED with different CCT and Ra values are illustrated for comparison. For the WLED with 3068 K, the portion of the blue hump is very limited, in comparison with that of the red hump. Meanwhile, for the WLED with 4932 K, the portions of blue, yellow, and red light are comparable, and it provides the highest Ra, of 87.4, in comparison with others. The highest value of the optimal Ra that we can achieve is mainly limited by the combination of humps produced by InP QDs. To further improve Ra, the species of QDs and the proportion of different monochromatic humps of the WLED should be optimized.

To facilitate discussion, we define three WLED samples with different combinations of InP QDs. Additional to the differences in peak wavelengths of yellow InP QDs, blue LED chips with peak wavelength of 460 nm (the hump 1 in Figure 1) and InP QDs with peak wavelength of 600 nm (the hump 5 in Figure 1) are used to emit blue and red light for these three samples. For samples 1, 2, and 3, we select yellow InP QDs with peak wavelengths of 513 nm, 527 nm, and 556 nm, respectively. Table 1 displays the optimized CCT and Ra values of samples 1, 2, and 3 under various target CCTs. When the target CCT changes, the optimized CCT can be achieved near the target CCT; however, the optimized Ra increases first and then decreases with the increasing target CCT value. The highest Ra is achieved with sample 1 with the target CCT of 6000 K. Since the gap between the yellow hump and red hump of sample 3 is much larger than that of sample 1, the highest Ra that can be achieved is much lower than that of sample 1.

We assume that the target CCT is 6000 K, initial value of *Φ*_0_ is 10 W, the light distribution of WLEDs is a standard normal distribution, the obtained illuminance of the window is 10,000 lux, the dimension of window is 2.4 m^2^, and the distance between neighboring LEDs is 2 m. *E_t_* of the WLED matrix is uniformly calculated as 468 lux, and is relatively higher than the average value of *E_r_* because of the amount of light emitted from neighboring WLEDs in *E_t_*, as shown in Figure 6a. In Figure 6b, we observe a uniform distribution of *E_o_* in the middle of the WLED matrix; however, *E_o_* decreases slightly at the edge of the WLED matrix.

From the chromatic projection plane of *E_o_* at the bottom of Figure 6b, we observe an axial symmetrical distribution of *e_o_* to the normal direction of the window; *e_o_* at the corner is lower than those in the middle of the WLED matrix, because the compensation of illuminance from sunlight at the corner is weaker than that of other positions. Despite the difference between *e_o_* at the corner and *e_o_* in the center, the uniformity of *E_o_* is considerable, because the largest gap between the highest and the lowest *e_o_* is within 2 lux. This difference is hard to detect with human eyes when users stand under the WLED matrix. In other words, *e_o_* is very close to the target value of *e_t_*. Calculation results presented in Figure 6c show a similar symmetrical property as that of Figure 6b, however, *e_r_* in the middle position is much lower than *e_r_* at the corner. This is because WLED units at the corner should produce more optical energy for users to maintain a rated *e_t_*, since they lack natural light and light from neighboring WLEDs. Additionally, the average value of *E_r_* is much lower than that of *E_t_* and *E_o_*, indicating that the consumed optical energy offered by WLEDs is lower than the optical energy received by users. In other words, we can save a large amount of optical energy with the proposed method.

Suppose that the amount of table top in use is dynamically changing, and only a portion of table tops are occupied during working hours. We randomly set some WLED units as “inactive”, and only open a portion of WLEDs. The target *e_t_* in Figure 7 is the same as that of Figure 6 for all working WLEDs. For those unoccupied table tops, the corresponding *e_t_* of WLEDs are only influenced by neighboring WLEDs. The distribution characteristics and orders of magnitude of *E_o_* and *E_r_* in Figure 7b,c are similar to that of Figure 6, indicating that turning off a portion of WLEDs will not influence the optimization results of the WLED matrix.

Assume the room size is large enough, the distribution of *E_o_* is influenced by the interval between WLEDs of the matrix (*L_int_*). Here, we set the window dimension as 2.4 m^2^ and the matrix size as 10 × 10. Figure 8a–d represent *E_o_* of the WLED matrix with *L_int_* of 2, 3, 5, and 10 m, respectively. The common characteristic of Figure 8a–d is that they both present symmetrical property with respect to the normal line of the window. When *L_int_* is small, *E_o_* presents a high value in the middle and a relatively lower value in the corner. With the increasing of *L_int_*, only the portion of *e_o_* that is near the window presents relatively high values. Those areas far from the window present a uniform distribution with the lowest *e_o_* of 365 lux, because these positions are almost unaffected by natural light and neighboring WLEDs. The movement of the window position is also considered to observe the distribution of *E_o_*. As shown in Figure 8e,f, the window position is shift to the left side in comparison with its original position with 2 and 6 m. The asymmetrical distribution of *E_o_* representing that the position of window cause a substantial influence to the illumination distribution uniformity of the WLED matrix.

To evaluate the energy saving ability of the proposed method, we calculate the energy saving percentage (*E_s_*) of the WLED matrix with dimensions ranging from 3 × 3 to 100 × 100 under different natural light illuminances. The natural light illuminances are selected as 2500 lux, 5000 lux, and 10,000 lux for comparison. From the left axis of Figure 9, we observe *E_s_* presents a drastic increasing trend when the WLED matrix is smaller than 20 × 20, and presents a decreasing trend when we continually increase the matrix dimension. This decreasing trend gradually becomes smooth when the matrix dimension continually increases. A similar phenomenon appears when the window provides different natural light illuminances. When natural light illuminance of the window increases to 10,000 lux, the highest *E_s_* can reach up to approximately 40%. It is obvious that the natural sunlight produces strong influence to the illumination energy consumption of the indoor area. *E_s_* decreases to almost 0.4% when the matrix increases to 100 × 100, which is an extreme condition. This is because the natural light produces large support to the WLED illumination system when the window dimension and the WLED matrix dimension are comparable. If the window dimension is much smaller than that of the WLED matrix, the optical energy from outside environment produces limited influence to the illumination of the WLED matrix.

When we increase the dimension of the WLED matrix, the running time of the program increases as well. From the right axis of Figure 9, we observe a mild increasing trend at first, and a following steep increasing tread when we increase the matrix dimension. With extreme condition that the matrix dimension reaches up to 100 × 100, the running time reaches approximately 95 s. These data only produce relative values since different operating systems provide different working speeds. When we use a WLED matrix with regular dimension, such as less than 20 × 20, the running time is less than 0.85 s, reflecting the rapid response of our method.

## 6. Conclusions

In this study, we propose an automatic illumination control method for an indoor InP-QD-based WLED matrix to minimize optical energy consumption by considering the influence caused by the outdoor environment and neighboring WLED units. We strategically realize an optical-spectrum-tunable WLED matrix by independently controlling the driving currents of different blue LED chips. With this method, Ra of the WLED system can be optimized under different target CCTs, which is also tunable according to the outdoor environment. We achieve a symmetrical distributed *E_o_* of illuminated table tops that are influenced by the distribution of WLEDs, natural light, and light from neighboring WLEDs. Additionally, the difference between *E_o_* and the target value of *E_t_* is negligible, indicating that all illuminated table tops possess uniform light distribution. The proposed automatic illumination control method demonstrates fast response times and shows promise for application in intelligent control LED systems in industrial workshops and office buildings to reduce energy consumption.

## Figures and Tables

**Figure 1 micromachines-13-01767-f001:**
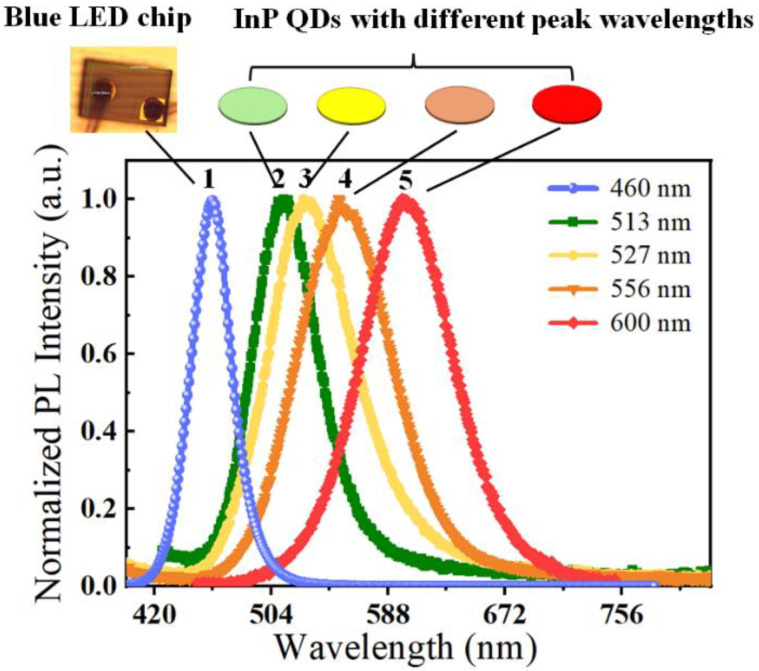
Monochromatic spectra produced by the blue LED chip and InP QDs with different peak wavelengths. The peak wavelengths of humps 1 to 5 are 460, 513, 527, 556, and 600 nm, respectively.

**Figure 2 micromachines-13-01767-f002:**
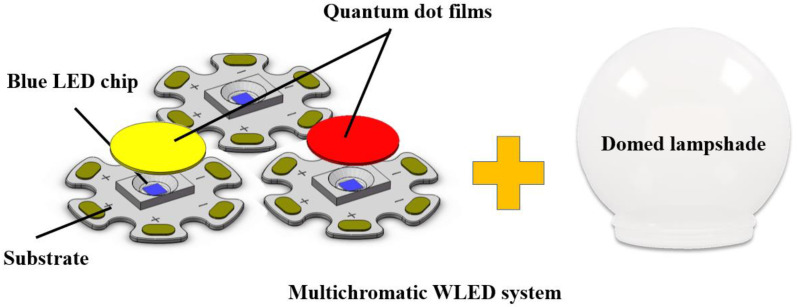
Assembly of the trichromatic WLED with three blue LED chips and two species of InP QDs with different peak wavelengths.

**Figure 3 micromachines-13-01767-f003:**
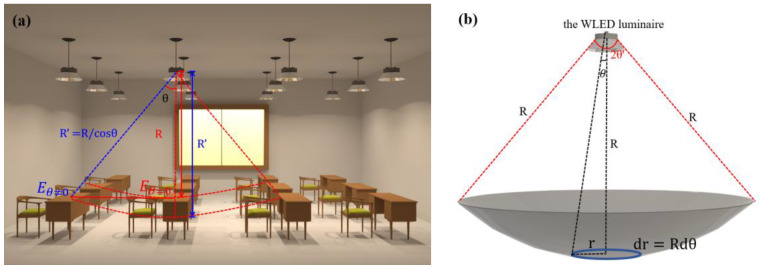
(**a**) Illumination scenario of the WLED matrix in the indoor area. (**b**) Schematic diagram of the transmission-distance-based optical distribution of a WLED luminaire.

**Figure 4 micromachines-13-01767-f004:**
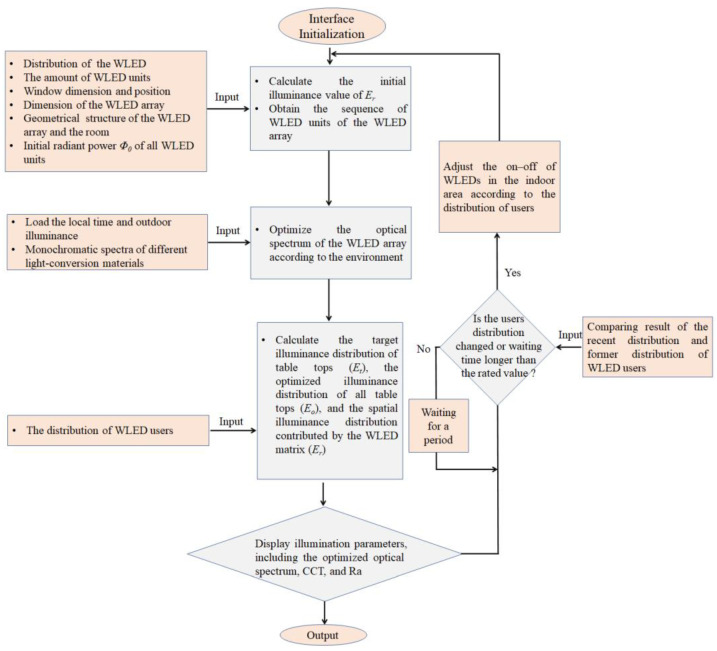
Flow diagram of the automatic illumination control program for the indoor WLED matrix.

**Figure 5 micromachines-13-01767-f005:**
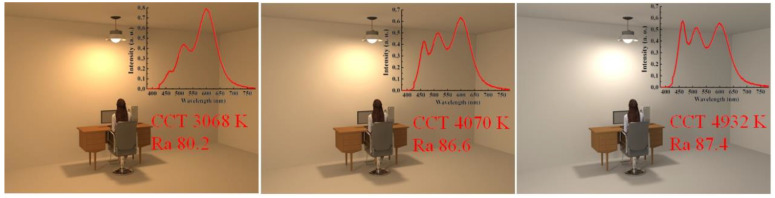
Illumination scenarios of the indoor area with WLEDs that produce different optical spectra, CCT, and Ra values.

**Figure 6 micromachines-13-01767-f006:**
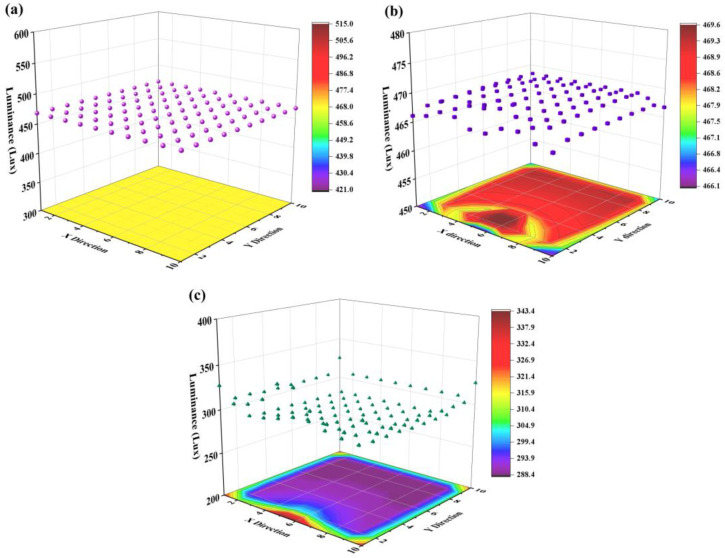
(**a**) Target spatial illuminance distribution (*E_t_*) of the WLED matrix. (**b**) Optimized spatial illuminance distribution (*E_o_*) of table tops under the WLED matrix. (**c**) Spatial illuminance distribution emitted by the WLED matrix (*E_r_*). The calculation results were obtained under the conditions that the dimension of the WLED matrix is 10 × 10, the error range (*δ*) is set as 2 lux, the illuminance of the window is 10,000 lux, the dimension of the window is 2.4 m^2^, and the distances between neighboring WLEDs are 2 m.

**Figure 7 micromachines-13-01767-f007:**
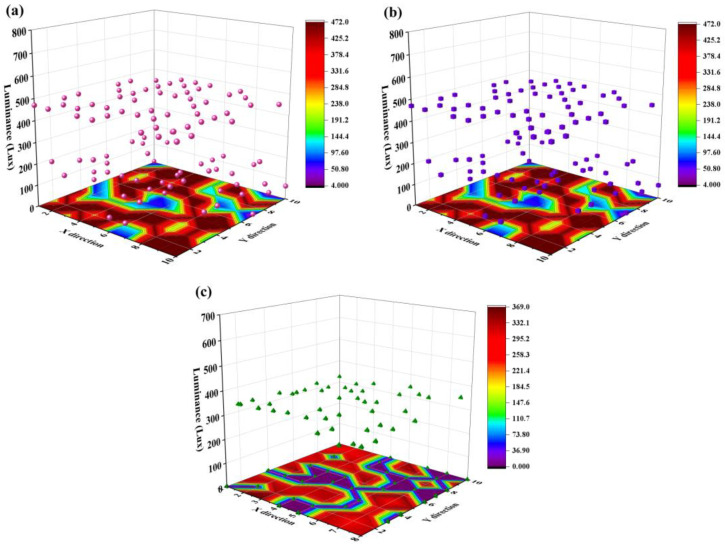
(**a**) Target spatial illuminance distribution (*E_t_*) of the WLED matrix with inhomogeneous distribution of users. (**b**) Optimized spatial illuminance distribution (*E_o_*) of table tops under the WLED matrix with inhomogeneous distribution of users. (**c**) Spatial illuminance distribution emitted by the WLED matrix (*E_r_*) with an inhomogeneous distribution of users. Calculation results were obtained under similar conditions as those in Figure 6.

**Figure 8 micromachines-13-01767-f008:**
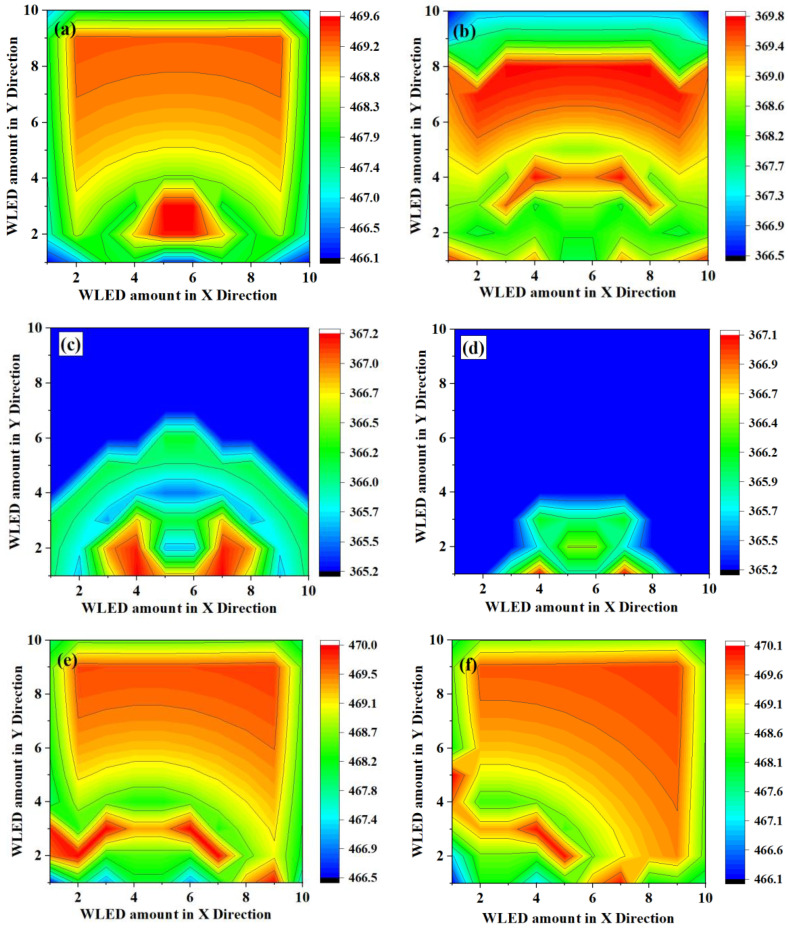
Optimized spatial illuminance distribution (*E_o_*) of the WLED matrix with (**a**) distance of 2 m between neighboring WLEDs; (**b**) distance of 3 m between neighboring WLEDs; (**c**) distance of 5 m between neighboring WLEDs; (**d**) distance of 10 m between neighboring WLEDs; (**e**) 2 m movement of the window position (distances of WLEDs are set as 2 m); (**f**) 6 m movement of the window position (distances of WLEDs are set as 2 m). Dimension of the WLED matrix equals 10 × 10.

**Figure 9 micromachines-13-01767-f009:**
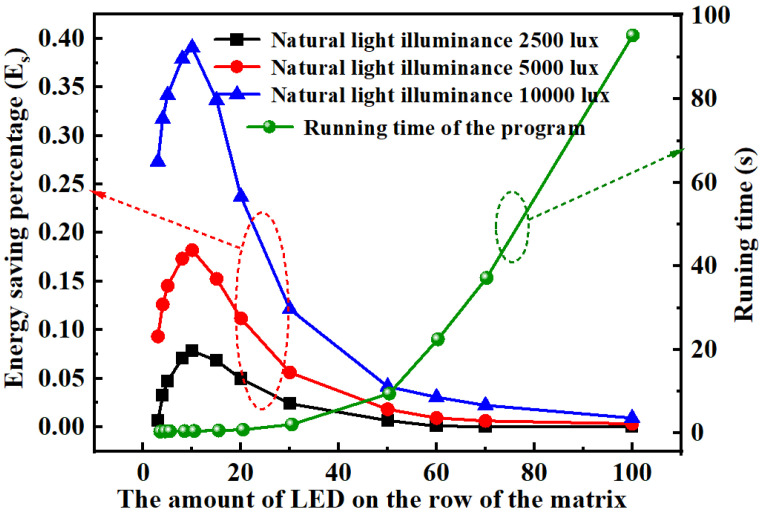
Energy saving percentage (*E_s_*) of the WLED matrix with different dimensions with natural-light illuminances of 2500 lux, 5000 lux, and 10,000 lux (left axis); the shifting trend of running time of our program with the increase in the matrix dimension (right axis).

**Table 1 micromachines-13-01767-t001:** Optimized CCT and Ra values of sample 1, 2, and 3 under various target CCT values.

Target CCT (K)	Sample 1 (Yellow InP QD with Peak Wavelength of 513 nm)		Sample 2 (Yellow InP QD with Peak Wavelength of 527 nm)		Sample 3 (Yellow InP QD with Peak Wavelength of 556 nm)	
	Optimized CCT (K)	Optimized Ra	Optimized CCT (K)	Optimized Ra	Optimized CCT (K)	Optimized Ra
3000	3068	80.2	3098	78.1	3096	74.8
4000	4070	86.6	4028	84.8	4001	81.4
5000	4932	87.4	4912	86.8	4994	83.9
6000	5927	87.5	6091	87.4	6105	75.7
7000	6951	87.2	7100	73.6	6960	72.5

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
