# Peer review of "Automatic Illumination Control Method for Indoor Luminaires Based on Multichromatic Quantum Dot Light-Emitting Diodes"

_micromachines, 2022, doi:10.3390/mi13101767_

Round 1

Reviewer 1 Report

This paper studied the illuminance distribution of InP QDs based WLEDs with considering the spectrum, number of units and distance amongst the WLEDs. There are several issues need to be addressed before consideration for publication.

1.       The authors emphasized the remaining issue that current illumination controlling studies seldomly simultaneously consider time-varying natural sunlight penetrating from the window and optical power originating from neighboring WLED units (line 82-83, page 2). However, there is no related study on this field. The authors individually studied the CCT of the WLEDs by altering the proportion of the yellow and red phosphors, the varying of sunlight intensity and CCT, and the spatial illuminance distribution by WLED distance and numbers. However, these are very basic design rules for commercial luminaire and in-door lighting system (Figure 6-7). Figure 5 provided an indoor lighting scene without sunlight. There is no example to combine these studies with considering the time-varying natural sunlight.

2.       The authors emphasized the light control method induce floating of spectra parameters (CCT and CRI) by accidentally increasing the fluctuation of the blue light. However, I did not see any necessity and advantage of using InP QDs because they resulted in low CRI and the authors did not provide the color coordinates of the WLEDs on the blackbody curve. Current commercial phosphors, such as LuAG, YAG, and KSF have high efficiency and color/ luminescence stability that are more suitable for this study.

Author Response

Please refer to the PDF, thanks.

Reviewer 2 Report

In this work, the authors propose a method for InP quantum-dots (QDs) based WLEDs to minimize optical energy consumption by considering the influence caused by outdoor environment and neighboring lighting WLED units. By dynamically control the light ingredients and optical power of WLEDs, the authors optimize the received illuminance distribution of table tops, improve lighting homogeneity of all users, and guarantee the lowest energy consumption of the WLED matrix. This work is of interest to other researchers in scientific and engineering community of LEDs. However, I do have some questions, which in my opinion should be addressed. The detailed comments are as follows:

1) In introduction, the authors write: “Light-emitting diodes (LEDs) have emerged as the fourth-generation solid-state lighting devices over the last decades due to their advantages of long lifespan, high efficiency, reliability, and eco-friendliness [1]. They have been widely applied in numerous scenarios, such as residential and outdoor illumination [2], rehabilitation therapy [3], visible light communication [4], agricultural planting [5], and indoor localization [6].” The general reference list seems a bit thin, considering the evolution in the field within the recent years. To give the readers a much broader view, recent developments related to GaN-based LED, such as Nano Energy 69, 104427 (2020); Optics Express 27(12), A669 (2019); Scientific Reports 8, 11053 (2018); Applied Physics Letters 118, 182102 (2021); Optics Letters 47(5), 1291-1294 (2022), etc. should be added, so that the readers can be clear about the state-of-the-art of this topic.

2) The WLED can be obtained by combination of blue LEDs and phosphor particles. What is the advantage of using InP quantum-dots?

3) The authors should provide the peak wavelength of blue-emissive LED chips.

4) In figure 6(b) and figure 6(c), the compensation of illuminance from sunlight is taken into consideration, leading to different illuminance distribution at the corner and that of other positions. Please explain why the compensation of illuminance from sunlight is not considered in figure 6(a)?

5) Figure 8 shows the spatial illuminance distribution of the WLED matrix with distances of 2 m, 3 m, 5 m, and 10 m between neighboring WLEDs, which means that the areas of these four rooms are different. There are two variables (the distance of the WLED matrix and the room area) in figure 8, the authors should keep one variable at a time.

6) In Fig. 1, “InP QDs with different peak wavelength” should be revised to be “InP QDs with different peak wavelengths”.

7) Some mistakes are found. For example, in figure 4, wrong spelling appears in “Distribution of the WLED The amount of WLED units”. 

8) In line 212, the authors initialized the window dimension and position. However, the dimension and position have an influence on the illuminance distribution. Can the authors investigate the influence of the dimension and position on the illuminance distribution?

Author Response

Please refer to the PDF, thanks.

Round 2

Reviewer 1 Report

The authors have answered the questions and the manuscript has been improved to satisfy the publication standard.